# Emerging Targeted Therapeutic Strategies to Overcome Imatinib Resistance of Gastrointestinal Stromal Tumors

**DOI:** 10.3390/ijms24076026

**Published:** 2023-03-23

**Authors:** Maria Teresa Masucci, Maria Letizia Motti, Michele Minopoli, Gioconda Di Carluccio, Maria Vincenza Carriero

**Affiliations:** 1Preclinical Models of Tumor Progression Unit, Istituto Nazionale Tumori IRCCS ‘Fondazione G. Pascale’, 80131 Naples, Italy; 2Department of Movement Sciences and Wellbeing, University “Parthenope”, 80133 Naples, Italy

**Keywords:** gastrointestinal stromal tumors, drug resistance, targeted therapies, biomarkers

## Abstract

Gastrointestinal stromal tumors (GISTs) are the most common malignant mesenchymal neoplasms of the gastrointestinal tract. The gold standard for the diagnosis of GISTs is morphologic analysis with an immunohistochemical evaluation plus genomic profiling to assess the mutational status of lesions. The majority of GISTs are driven by gain-of-function mutations in the proto-oncogene c-*KIT* encoding the tyrosine kinase receptor (TKR) known as KIT and in the platelet-derived growth factor-alpha receptor (*PDGFRA*) genes. Approved therapeutics are orally available as tyrosine kinase inhibitors (TKIs) targeting KIT and/or PDGFRA oncogenic activation. Among these, *imatinib* has changed the management of patients with unresectable or metastatic GISTs, improving their survival time and delaying disease progression. Nevertheless, the majority of patients with GISTs experience disease progression after 2–3 years of imatinib therapy due to the development of secondary *KIT* mutations. Today, based on the identification of new driving oncogenic mutations, targeted therapy and precision medicine are regarded as the new frontiers for GISTs. This article reviews the most important mutations in GISTs and highlights their importance in the current understanding and treatment options of GISTs, with an emphasis on the most recent clinical trials.

## 1. Introduction

Gastrointestinal stromal tumors (GISTs) are soft tissue sarcomas, representing the most common nonepithelial tumors of the gastrointestinal tract. GISTs originate from precursors of the interstitial cells of Cajal, the pacemaker cells, whose function is to signal the muscles in the gastrointestinal tract to contract for moving food and liquids [1,2]. GISTs can start anywhere along the gastrointestinal tract, with 50% of them being localized in the stomach, and the others mainly in the small intestine. However, GISTs can also localize outside the digestive tract, including in the omentum or retroperitoneum [3]. GISTs are common in older patients: the average age at diagnosis ranges from 62 to 75 years, with a peak incidence in the 8th decade of life. Less than 10% of patients are younger than forty years, whereas GISTs are quite rare in children and young adults [4]. Morphologically, GISTs may range in size from a diameter of a few millimeters to large masses measuring more than 30 cm and can appear as a single, well-circumscribed submucosal or polypoid mass [5]. During the past three decades, there has been a great debate regarding GIST nomenclature, cellular origin, and diagnosis. Due to their relatively broad morphologic spectrum, GISTs were classified as leiomyomas, leiomyosarcomas, and leiomyoblastomas of the gastrointestinal tract, until they were found to have clinical, histopathological, and molecular biological features differentiating them from other soft tissue sarcomas [6]. Based on cytology, GISTs are classified into three groups: spindle cell (70%), epithelioid (20%), and mixed spindle and epithelioid cell type (10%). Spindle cell GISTs are mainly composed of fusiform, spindle-shaped mesenchymal cells with indistinct cell borders organized in short intersecting fascicles. In epithelioid GISTs, tumor cells tend to exhibit a nested pattern of growth, showing an epithelial-like morphology with round-ovoid nuclei [7].

Almost 95% of GISTs express the c-KIT (CD117) tyrosine kinase receptor (TKR), encoded by the *c*-*KIT* proto-oncogene [5]. In the remaining cases in which tumor cells are weakly or barely positive for CD117, the histologic diagnosis of GISTs is more challenging and the presence/absence of other markers has to be evaluated. For instance, besides CD117, the majority of spindle cell type GISTs express DOG1, a voltage-gated transmembrane calcium-activated chloride channel [8]. About 90% of GISTs are driven by gain-of-function mutations in the TKR c-*KIT* or platelet-derived growth factor receptor-α (*PDGFRA*) causing the ligand-independent continuous kinase activation of the RAS–RAF–MAPK pathway, which results in cellular proliferation and oncogenesis [9,10]. In this regard, it is to be pinpointed that 10% of GISTs which are devoid of either *KIT* or *PDGFRA* mutations and referred to as wild-type GISTs (WT-GISTs) are much more commonly associated with younger patients [11,12,13]. For subtype WT-GISTs, the immunohistochemical evaluation of subunit B of succinate dehydrogenase (SDHB), an enzyme involved in the mitochondrial electron transport chain, is recommended in order to separate the *SDH*-competent from *SDH*-deficient GISTs. SDH-deficient variants are the most frequent WT-GISTs, and the loss of SDH activity seems to play a key role in the pathogenesis of these tumors [8,13,14]. Knowledge of the molecular subtype has important implications for therapeutic strategies: in SDH-deficient GISTs, tumor, and germline SDH-complex genes (*A, B, C, or D*) sequencing should be performed; in tumors lacking the SDH mutation, the presence or absence of SDHC promoter methylation should be assessed. Instead, in the competent SDH subtype, mutations in other molecular target genes could predict response to specific targeted therapies [15,16,17] (Figure 1).

Moreover, the possible presence of gene rearrangements could guide therapy, as for neurotrophic tyrosine receptor kinase gene fusions (NTRK) harboring GIST and treatable with NTRK inhibitors [18,19,20]. Based on these considerations, both the National Comprehensive Cancer Network (NCCN) and the European Society for Medical Oncology (ESMO) recommended the genotyping of GISTs for choosing the best therapeutic strategy both in the adjuvant and metastatic setting [19,21]. Importantly, since the development of GISTs is quite often related to proliferative signals associated with germline mutations, it is necessary to genetically analyze both tumor and germline cells [19,21] (Figure 1). Today, next-generation sequencing (NGS) technologies are applied to simultaneously and efficiently identify GIST-related gene mutations and provide useful references for clinicians. 

In this review, we focus on GIST molecular subtypes acting both as therapeutic targets and causing resistance to targeted therapy. Moreover, current treatments for GISTs and the most recent clinical trials are discussed.

## 2. KIT Mutations in GISTs

For the first time, by sequencing the *c-KIT* complementary DNA from five GISTs, Hirota and coworkers found that mutations in the KIT region, between the transmembrane and tyrosine kinase domains, contribute to GIST development [9]. More than 90% of GISTs express the 145 kDa transmembrane glycoprotein c-KIT (CD117) encoded by the proto-oncogene c*-KIT*. Among these, 70–85% harbor a *KIT* mutation, conferring a gain of function [9,22,23], which results in the constitutive activation of the protein, leading to GIST tumorigenesis [9,24,25,26]. CD117 belongs to the type III TKR family and is constituted by an extracellular domain (exons 1–9), a transmembrane domain (exon 10), a juxtamembrane domain (exon 11), and a tyrosine kinase domain (exons 13–21), and is encoded by the human c-*KIT* proto-oncogene located on chromosome 4q12 [27]; stem cell factor (SCF) is the KIT-specific ligand [21,28,29]. Upon binding to its ligand, KIT dimerization, and activation result in a downstream signaling cascade, which involves JAK–STAT3, phosphatidylinositide-3-kinase (PI3K)–AKT–mTOR, and RAS–MAPK pathways, thus regulating cell proliferation and apoptosis, chemotaxis, and cell adhesion [30,31,32]. The most common mutations arise in the juxtamembrane domain encoded by exon 11 followed by mutations in the extracellular domain, often due to duplications in exon 9, mutations in exons 13 and 14 that affect the ATP-binding pocket, and mutations in exon 17 located in the true kinase domain. 

Mutations in exon 11, consisting of a deletion (del V557-D558), disrupt the autoinhibitory domain of the receptor, thus allowing continuous kinase activation in 70–75% of GISTs [9,33,34]. A duplication in exon 9 (AY502-503 dup) represents the second most common mutation in GISTs and accounts for 10–15% of newly diagnosed GISTs [35].

These mutations occur more commonly in GISTs arising from the small intestine [23,34]. GISTs mutated in exons 9 and 11 do respond to imatinib, but at a different rate: 90% of patients with a *KIT* exon 11 mutation are sensitive to imatinib, whereas only 50% of patients with a *KIT* exon 9 mutation respond since a steric modification of the extracellular domain of KIT molecule occurs, hampering imatinib binding [36,37]. In these last patients, the drug therapeutic dosage must be increased to ameliorate the clinical response [38,39]. Mutations in exon 13 (K642E) encoding the ATP-binding region of KIT interfere with the physiological autoinhibitory function of the juxtamembrane domain and usually arise in the stomach. Other primary mutations occur in exon 17, the most common being N822K. They are rare and account for approximately 1% of newly diagnosed GISTs [40,41]. The exon 17 secondary mutations arise more frequently in GISTs of the small intestine compared to those of the stomach and involve codons 816, 820, or 823 [40].

Finally, rare mutations of the *c-KIT* gene in exon 8 that make the receptor constitutively active, as it is capable of autophosphorylation, have been reported for the first time in familial cases with germline mutation of the gene leading to Del-Asp419. Subsequently, cases of GISTs with substitution of ThrTyrAsp (417–419) to Tyr (TYD417-419 Y) were found. However, the number of GIST cases with exon 8 mutations appears to be very small [42]. The most frequent *KIT* gene mutations in GISTs are shown in (Figure 2). 

## 3. PDGFRA Mutations in GISTs

Activating mutations of *PDGFRA,* another member of the type III TKR family has been identified in 10–15% GISTs, which is functionally and structurally homologous to KIT [23,43,44,45]. GISTs harboring *PDGFRA* mutations are essentially different from those harboring *KIT* mutations and occur in different sites, being the *PDGFRA-*mutant GISTs characteristic of the stomach or the omentum, with rare cases originating in the intestine or mesentery [34,46,47]. In epithelioid and mixed variants of GISTs, the expression of *KIT/PDGFRA*-mutant isoforms is associated with the anatomical site of the tumor [47]. 

Like *KIT* mutations, *PDGFRA* mutations confer a gain of function as they disrupt the TKR autoinhibitory regions, thereby resulting in a ligand-independent activation [10,48]. *PDGFRA* mutations occur in about 5–7% of GISTs, involving the A-loop encoded by exon 18 (~5%), rarely the juxtamembrane region encoded by exon 12 (~1%), or the ATP-binding domain encoded by exon 14 (<1%) [49]. The most common *PDGFRA* mutation is the D842V point mutation, located within the kinase domain activation loop (encoded by exon 18) [48,50]. Substitution at codon D842 in exon 18 constitutes 63% while the deletions p.D842_H845 (DIMH842-845) and p.I843_D846 (IMH843-846) account for 15%. Corless and colleagues demonstrated that CHO cells stably transfected with *PDGFRA* mutants involving the codon D842 in exon 18 (D842V, RD841-842KI, and DI842-843IM) are resistant to imatinib, except for the D842Y that is sensitive. On the other hand, the mutations D846Y, N848K, Y849K, and HDSN845-848P in exon 18 show sensitivity to imatinib, in vitro [48]. Other *PDGFRA* mutations include the ATP-binding pocket encoded by exon 14 (N659K), which is homologous to *KIT* in exon 13, and exon 12 (SPDHE566-571R and insertion ER561-562) [48]. Almost all *PDGFRA* mutations do not give resistance to TKIs, except the point mutation D842V which confers resistance to imatinib by preventing imatinib binding to the ATP-binding site in about 10% of primary GISTs [33,35,39]. Importantly, the survival outcomes of patients with advanced-stage *PDGFRA*-mutant GISTs are poor because *PDGFRA* D842V-mutant GISTs are highly resistant to imatinib. In a cohort of 58 patients with P*DGFRA*-mutant GISTs treated with imatinib, no clinical response was elicited in patients with a *PDGFRA* D842V mutation [45]. The most frequent *PDGFRA* gene mutations in GISTs are shown in (Figure 2). The sensitivity to TKI of KIT and PDGFRA-mutated GISTs is shown in Table 1. 

## 4. Other Mutations in GISTs

Approximately 15% of adult patients with GISTs are negative for mutations in *KIT* or *PDGFRA* genes. These so-called WT-GISTs are characterized by other oncogenic drivers, including mutations in the subunits of the succinate dehydrogenase (*SDH*) complex, in the serine-threonine protein kinase BRAF *(BRAF)*, Neurofibromatosis type 1 (*NF1),* or the neurotrophic tyrosine receptor kinase (*NTRK*) genes (Figure 1). 

### 4.1. SDH Mutations

Almost 50% of KIT and PDGFRA WT-GISTs are marked by alterations involving the SDH complex [7]. The *SDHA, SDHB, SDHC,* and *SDHD* genes encode four subunits of the succinate dehydrogenase mitochondrial complex, an enzyme anchored to the inner mitochondrial membrane and involved in energy-producing metabolic processes. SDHA converts succinate into fumarate, SDHB participates in the electron transport chain to oxidation of ubiquinone to ubiquinol, whereas SDHC and SDHD are membrane-anchoring subunits [17]. The loss of any SDH subunit renders the complex inactive and leads to a loss of SDHB detectable by immunohistochemistry. GISTs lacking *KIT* or *PDGFRA* mutations can be SDH-competent or SDH-deficient. Their SDH status should be determined since some SDH-competent GISTs are aggressive and tend to metastasize, whereas SDH-deficient tumors are characterized by an indolent overall clinical course and longer OS, although they do not respond to systemic therapies [17]. SDH-competent tumors occur in older patients and 82% in the small bowel, whereas SDH-deficient tumors originate from the stomach.

The SDH-competent GIST group includes a large proportion of patients who may harbor either *KIT* or *PDGFRA* mutations and also mutations in genes involved in the RAS–MEK–MAPK pathway, and translocations involving *NTRK* or *FGFR* genes [7,51]. The knowledge of the molecular subtype in these patients has important implications for therapeutic strategies since the occurrence of kinase mutations can predict the responsiveness to targeted therapies [15,16,17]. 

SDH-deficient GISTs comprise a subgroup of relatively rare tumors that lose the SDH complex, either by the combination of somatic and germline mutations in the *SDH*-subunit genes or by epigenetic mechanisms [17,52,53]. Germline mutations in *SDHA* occur in approximately 30% of the SDH-deficient GISTs, whereas those in *SDHB*, *SDHC*, and *SDHD* are less frequent [54]. *SDH*-deficient variants are the most frequent WT-GISTs, and the loss of SDH activity seems to play a key role in the pathogenesis of these tumors [13]. However, not all SDH-negative GISTs harbor an *SDH* gene mutation, since these tumors may have other epigenetic and genetic defects in the SDH pathway [55].

Clinically, SDH-deficient GISTs are restricted to the stomach, occur predominantly at a young age, and respond poorly or are resistant to imatinib. Mortality is almost 15%, although the behavior of these tumors is unpredictable since metastases may develop after a long time [52,56]. In SDH-deficient GISTs, the overexpression of the insulin-like growth factor 1 receptor (IGF1R) has been reported, suggesting a potential role of IGF1R as a target for inhibition therapy [13]. Recently, in a phase II trial, an oral small-molecule, vandetanib (ZD6474), an inhibitor of VEGFR2, EGFR, and RET has been evaluated in patients with SDH-deficient GISTs (dSDH GISTs). Unfortunately, no partial or complete responses have been obtained and the authors concluded that vandetanib is neither effective nor well tolerated in these patients [57]. 

### 4.2. BRAF Mutations

BRAF is a member of the RAS–RAF–MEK pathway involved in cell cycle regulation and the oncogenic modulation of cellular responses to growth signals via MAPK pathway activation [58]. The occurrence of the *BRAF* (V600E) mutation was initially described by Agaram and colleagues in subsets of *KIT/PDGFRA* wild-type and imatinib-resistant GISTs [59]. In GISTs, the *BRAF* gene is activated by somatic point mutation clustered in the kinase domain due to a single nucleotide substitution in exon 15 at codon 1799 [60,61]. Initially, *BRAF* and *KIT/PDGFRA* mutations were considered mutually exclusive, representing 3.5–13.5% of primary wild-type *KIT/PDGFRA* GISTs. In a cohort of 172 wild-type *KIT/PDGFRA* GISTs, Huss and coworkers found *BRAF* mutations in only 3.9% of patients [62]. More recently, several studies revealed that the *BRAF* (V600E) mutation could occur in 2% of GISTs carrying mutated *KIT/PDGFRA* with acquired resistance to imatinib [63], highlighting the possibility that the frequency of *BRAF* coexistence with *KIT/PDGFRA* mutations was under-estimated in past years due to the lack of highly sensitive analytical methods [64]. However, more recently, using a quantitative competitive allele-specific Taq-Man duplex PCR, Jašek and colleagues confirmed the concomitant but rare occurrence of *BRAF/KIT* and *BRAF/PDGFRA* mutations in GISTs [64]. Accordingly, Torrence D. and colleagues have reported two spindle cell phenotype GIST cases harboring novel *BRAF* fusion genes arising in two young-adult women in the small bowel and esophagus. In both cases, immunohistochemical analysis revealed a diffuse reactivity for DOG1, while KIT/CD117 was weakly positive or negative. Conversely, targeted RNA sequencing with Archer Fusion Plex revealed the occurrence of a fusion between *BRAF* with either *AGAP3* or *MKRN1* gene partners [65]. These findings attest to the importance of the *BRAF* (V600E) mutation as an emerging, uncommon but established oncogenic driver in GISTs, whose role as a target marker for TKIs needs to be further investigated. 

### 4.3. NF1 Mutations

Neurofibromatosis type 1 is an inherited genetic disorder characterized by cancer predisposition due to a mutation in the neurofibromin type 1 (*NF1)* gene coding for the neurofibromin, a tumor suppressor protein [66]. NF1 is a negative regulator of RAS; being a GTPase activating protein (GAP) it promotes GTP hydrolysis. Thus, NF1 loss activates RAS and, in turn, RAF/MEK/ERK [67]. Furthermore, RAS-GTP interacts with the p110α catalytic subunit of phosphoinositide 3-kinase (PI3K), which converts PIP2 to PIP3, recruiting and activating AKT and, consequently, the AKT/mTOR signaling pathway [68]. Somatic or germline *NF1* mutations result in a loss of function of neurofibromin, leading to increased proliferation and an increased risk of developing malignancies, including GISTs [69,70,71,72]. An increasing number of studies reported the association between *NF1* and GISTs occurring more frequently in females, with very variable incidence, estimated between 3.9% and 25% [73,74]. Most of *NF1*-associated GISTs are constituted by spindle cells [75], and their preferred localization is at the transition of the duodenum into the jejunum [76]. It has been reported that *NF1-*related GISTs refer to a subset of WT-GISTs, probably implying a different molecular pathogenesis as they only occasionally express CD117 [11,69,77,78,79]. Since most GISTs occurring in patients with *NF1* have no activating mutations in *KIT*, *PDGFRA,* or *BRAF* [80,81,82], these tumors respond poorly to imatinib or other TKIs, including sunitinib and regorafenib [79,80]. Recently, the MEK inhibitor selumetinib approved for the treatment of young patients affected by NF1 has been introduced in a phase II study registered on ClinicalTrials.gov, id: NCT03109301, which involves adult patients with *NF1*-mutated GISTs [13,83]. 

### 4.4. NTRK Mutations

The *NTRK* family consists of *NTRK1*, *NTRK2*, and *NTRK3* genes encoding the tropomyosin receptor kinase (TRK) A, B, and C, respectively. Oncogenic TRK activation is mainly due to the fusion of *NTRK* genes and is involved in the pathogenesis of many tumors, including *WT*-GISTs [7,84,85,86]. GISTs with *NTRK* rearrangements occur less frequently in the stomach are frequently larger, and the epithelioid type has a higher risk of recurrence [87]. The ETS variant transcription factor 6 (*ETV6-NTRK3*) fusion is an actionable target in GISTs: Brenca and colleagues found that ETV6-NTRK3 might trigger the insulin-like growth factor-1 receptor-signaling cascade and the alternative nuclear insulin receptor substrate-1 pathway to promote the development of GISTs [88]. Eight GISTs harboring *NTRK1* or *NTRK3* rearrangements have been described by Atiq M. and collogues. These cases were morphologically heterogeneous and showed variable clinical outcomes. A diffuse pan-TRK expression was demonstrated by immunohistochemistry, and the molecular genetic analysis revealed the occurrence of 3 *TPM3-NTRK1*, *TPR-NTRK1*, *LMNA-NTRK1*, and 2 *ETV6-NTRK3*, *SPECC1L-NTRK3* in-frame gene fusions [89]. However, among the techniques applied to identify *NTRK* fusions, the use of NGS and RNA sequencing performs with strong consistency, while immunohistochemical staining for Pan-TRK or FISH has limited specificity [90]. Among the *NTRK* fusions, the NTRK3 fusion is more common than the NTRK1 and NTRK2 fusions [88]. In quadruple-negative GISTs, the occurrence of the *ETV6-NTRK3* gene fusions has been identified by a massive parallel sequencing approach [88].

Based on these considerations, there is increasing interest in introducing TRK inhibitors for the treatment of GISTs with *NTRK* fusions. A phase II study of the TRK inhibitor larotrectinib enrolled 55 patients with TRK-fusion-positive cancers. Three patients had TRK-fusion-positive GISTs, two of which achieved a partial response and one achieved a complete response [91]. Similar to larotrectinib, the TRK inhibitor entrectinib has been approved by the United States Food and Drug Administration (FDA) for the treatment of solid tumors, including some soft tissue sarcomas with *NTRK* fusions. These TRK inhibitors exhibited antitumor efficacies in a variety of tumors harboring *NTRK* fusions, including WT-GISTs, and phase I and II clinical trials for larotrectinib and entrectinib (Id: NCT02576431 and Id: NCT02568267, respectively,) are currently ongoing [91,92,93,94,95,96].

## 5. Imatinib for the First-Line Therapy of GISTs 

Imatinib mesylate (STI571) is an orally available tyrosine kinase inhibitor (TKI) of KIT and PDGFRA receptors initially developed to treat chronic myeloid leukemia by inhibiting the intracellular kinases ABL and BCR-ABL fusion protein [97]. The drug can block the transfer of phosphate groups from adenosine triphosphate to tyrosine residues of the substrates, hence interrupting the downstream signaling cascade that regulates cell proliferation. Imatinib has been the first targeted therapy approved for the treatment of GISTs, representing adjuvant and neoadjuvant first-line therapy in CD117-positive advanced GISTs [98,99,100].

GISTs can be sensitive or insensitive to imatinib. Two types of resistance to imatinib may occur. Primary resistance, depending on specific tumor genotypes referred to as primary mutations, and secondary resistance is related to the development of new mutations (i.e., secondary mutations) raised during the therapy that had been initially able to control GIST progression [39]. Of note, patients are considered primarily resistant to TKIs also in the case of tumor progression within the first six months of treatment [101,102,103]. 

Secondary *KIT* mutations in GISTs are commonly identified in post-imatinib biopsy specimens of patients who underwent first-line therapy with imatinib and should be related to the selective pressure exerted on tumor cells by the primary treatment [104,105]. *KIT* secondary mutations are generally located in exons 13, 14, 17, and 18. Indeed, several mutations could appear concomitantly. Based on the sensitivity of the method, secondary mutations have been found in 44–90% of GISTs harboring primary mutations [106]. The acquired mutations are mainly located in the ATP-binding pocket (exon 13—V654A mutation—and exon 14—T607I mutation) and the activation loop (exons 17 and 18) of the tyrosine kinase domain [107,108,109] (Figure 2). These mutations reduce or prevent imatinib binding, by disrupting H-bonds or modifying the conformation of the protein, thus making the tumor resistant to imatinib first-line therapy [110,111]. 

## 6. Sunitinib for Second-Line Therapy of Resistant GISTs

In clinical practice, to overcome secondary imatinib resistance, other TKIs may be used since resistant GISTs may still be ontogenically related to the activation of KIT downstream signaling for cell survival and proliferation [104] (table in below).

Sunitinib malate, a multitargeted TKI, targeting KIT, PDGFRA, VEGFR, and several other kinases is regarded as the standard second-line therapy for secondary resistant GISTs [112,113,114]. Sunitinib binds to the ATP-binding pocket of the inactivated KIT, blocking its autoactivation [115]. The clinical safety and activity of sunitinib have been documented in several clinical studies. In patients affected by advanced GISTs with secondary resistance to imatinib, Demetri and colleagues conducted a randomized, double-blind, placebo-controlled, multicenter trial (ClinicalTrials.gov, id: NCT00075218) to evaluate the tolerability and anticancer efficacy of sunitinib, being the time of tumor progression the primary endpoint. In this study, sunitinib was well tolerated, and able to control tumor progression. Of note, the median time to tumor progression was 27.3 weeks in patients receiving sunitinib, whereas it was only 6.4 weeks in patients receiving the placebo [116]. Heinrich and coworkers assessed that the clinical activity of sunitinib after imatinib failure is highly influenced by primary and secondary mutations in the most common pathogenic kinases, having implications for the optimization of the treatment. The impact of primary and secondary kinase genotype on sunitinib activity was investigated in 97 patients with metastatic, imatinib-resistant GISTs, in the phase I/II trial. The clinical benefit of sunitinib was observed for the most frequent primary GIST genotypes: *KIT* exon 9 (58%), *KIT* exon 11 (34%), and wild-type *KIT/PDGFRA* (56%). PFS was longer for patients with the primary *KIT* exon 9 mutation [109]. In an open-label, multicenter, phase II trial (clinicaltrials.gov id: NCT00137449), George and coworkers investigated a different scheme of sunitinib administration to ameliorate safety and tolerance. The study recruited 60 GIST patients not amenable to standard therapy and with documented resistance or intolerance to imatinib. The primary endpoints were clinical benefit rate, partial responses, and stable disease (>24 weeks). Secondary endpoints included PFS, OS, safety, pharmacokinetic parameters, and plasma biomarker levels. The study established that continuous daily oral administration of 37.5 mg sunitinib for 28 days is an active alternative dosing strategy with acceptable safety [117]. Interestingly, the response rate to sunitinib depends on the site of the mutations. Indeed, patients harboring *KIT* mutations in exon 9 respond better than those with mutations in *KIT* exon 11 [118]. In particular, GISTs carrying *KITAY502-3* mutations at exon 9 exhibit the highest sensitivity to sunitinib [119]. The response rate to sunitinib was higher in patients harboring secondary *KIT* mutations in exon 13 or 14 compared to secondary *KIT* mutations in exon 17 or 18, with better PSF and OS [109]. In vitro, sunitinib is more active on *KIT* GIST cell lines bearing secondary mutations at the ATP-binding pocket (exons 13 and 14) than in GIST cell lines harboring imatinib-resistant mutations at the activation loop (exons 17, mutations D820Y, D820E, and N822K, and exon 18 A829P) [109,120]. In preclinical studies, resistance to sunitinib has been demonstrated due to the presence of new mutations in the *KIT* activation loop (mainly in exon 17, for instance, *D816V*, *D816F,* and *T670I*) [119,121]. In the case of imatinib or both imatinib and sunitinib-resistant GISTs, vatalanib, a TKI of KIT, PDGFRA, and VEGFR has been tested in a phase II trial and demonstrated effectively [122]. 

## 7. Third-Line Therapy for Resistant GISTs

GISTs resistant to sunitinib and in tumor progression can be treated with other TKIs [123] (table in below).

### 7.1. Regorafenib

Regorafenib is an orally active multitarget TKI with antiangiogenic activity, effective against several solid tumors, including GISTs [124,125,126], with a manageable toxicity profile also in prolonged treatment [127]. On February 2013, regorafenib was approved by the FDA for use in the third-line setting for advanced GISTs after the failure/intolerance of imatinib and sunitinib. Preclinical studies demonstrated that regorafenib is active on some GISTs harboring secondary mutations and resistant to sunitinib [128]. The GRID trial “Study of regorafenib as a 3rd-line or beyond treatment for GISTs” (ClinicalTrials.gov Id: NCT01271712), is a randomized, double-blind, placebo-controlled phase III study, comparing the efficacy of regorafenib plus the best supportive care vs. placebo plus the best supportive care for patients with metastatic and/or unresectable GISTs whose disease progressed despite prior treatment with at least imatinib and sunitinib. The study enrolled 199 patients, the last update being on 29 January 2021. The study shows a PFS amelioration of patients treated with regorafenib compared with those receiving a placebo and was the first clinical trial to demonstrate a clear benefit of the use of a kinase inhibitor in this highly refractory patient population [129]. Regorafenib showed good efficacy and a manageable safety profile also in Japanese patients enrolled in the GRID study, although some side effects were more frequent than in western subjects [130]. Interestingly, a recent phase II study (clinical trial. gov Id: UMIN000016115) has ascertained that regorafenib is also active in the second-line therapy in patients with imatinib-resistant GISTs and that a secondary mutation in *KIT* can be predictive of the efficacy of regorafenib [128].

### 7.2. Nilotinib

Nilotinib is a TKI orally bioavailable amino-pyrimidine-derivative, able to overcome imatinib resistance in GISTs by inhibiting either *c-KIT* or *PDGFRA*. Nilotinib elicits a 20-fold higher efficacy on cell proliferation, in vitro, compared to imatinib or sunitinib [131]. In a retrospective analysis, Montemurro and collaborators reported the tolerability and effectiveness of nilotinib administrated within a compassionate program in GISTs in which treatment with both imatinib and sunitinib was ineffective, contraindicated, or not tolerated [132]. In the “Phase 2 study of nilotinib as a third-line therapy for patients with GISTs”, Akira Sawaki and colleagues analyzed the efficacy and safety of nilotinib in GIST patients resistant either to imatinib or sunitinib. Thirty-five patients were enrolled and treated with 400 mg nilotinib twice a day. The primary endpoint was disease control rate (i.e., the percentage of patients with complete response, PR, or SD lasting for 24 weeks or longer). The disease control rate at week 24 was 29%, and the median PFS and OS were 113 days and 310 days, respectively. Partial response was observed in only 3% of patients. Interestingly, one of these patients had a GIST harboring a *KIT* mutation at exon 11 and an imatinib-resistant and sunitinib-resistant *KIT* mutation on exon 17. Moreover, 66% of patients had an SD of ≥ 6 weeks. Thus, according to these results, the authors refer that nilotinib is well tolerated and retains antitumor activity in patients with imatinib- and sunitinib-resistant GISTs [133]. In the phase III study of nilotinib versus best supportive care with or without a TKI in patients with GIST resistance or intolerance to imatinib and sunitinib, nilotinib treatment gave a longer median OS [134]. In untreated GIST patients or patients with GIST recurrence after adjuvant imatinib therapy and not submitted to subsequent treatments, Casali and coworkers evaluated the efficacy of nilotinib as a first-line therapy. In the phase II trial: “Treatment of patients with metastatic or unresectable GISTs in first-line therapy with nilotinib” (ClinicalTrials.gov Id: NCT00756509), nilotinib exerted relevant clinical benefits, showing a good safety profile [135]. The “Phase II study aiming to evaluate the efficacy and safety of nilotinib in patients with GISTs resistant or intolerant to imatinib and or to 2nd-line TKI” (ClinicalTrials.gov Id: NCT00633295), last update posted on 22 June 2017, is still ongoing and is evaluating the efficacy of nilotinib by tumor uptake of FDG in PET at 6 months. Safety and tolerability are measured by the rate and severity of adverse events.

### 7.3. Pazopanib

Pazopanib is a broad-spectrum TKI, targeting KIT, PDGFRA, VEGFR1, VEGFR2, and VEGFR3. A phase II randomized multicenter study (ClinicalTrials.gov Id: NCT01323400) evaluating the “Efficacy of pazopanib + best supportive care (BSC) vs. BSC alone in metastatic and/or locally advanced unresectable GISTs resistant to imatinib and sunitinib”, was performed. In this study, pazopanib plus BSC was demonstrated to improve progression-free survival in patients with advanced GISTs resistant to imatinib and sunitinib, compared to BSC alone. Data from this trial provide input for subsequent studies of targeted inhibitors in the third-line setting for GIST patients [136]. Another study showed that pazopanib is well tolerated and rather effective, suggesting that it can be considered as a treatment option in advanced GISTs resistant to imatinib [137]. The PAGIST trial confirmed the results from the PAZOGIST trial. Indeed, in the third line, pazopanib is effective in almost 50% of patients with metastatic or locally advanced GISTs [138]. These results are similar to those obtained with regorafenib in third-line treatment [138]. 

### 7.4. Sorafenib

Sorafenib is a small molecular weight inhibitor of RAF kinase, PDGFRA, VEGFR2, VEGFR3, and c-KIT, simultaneously targeting the RAF/MEK/ERK pathway. It is employed for the treatment of patients with unresectable GISTs who failed previous standard treatments. In these patients, sorafenib controlled the disease for more than 24 weeks [139]. In a retrospective study, Montemurro and colleagues, after evaluating sorafenib efficacy in GISTs resistant to imatinib, sunitinib, and nilotinib, concluded that sorafenib is the most effective drug [140]. In a recent case report, Brinch and colleagues describe an advanced GIST patient, 61 years old, intolerant to imatinib, and progressed after sunitinib and nilotinib treatment. In this patient, sorafenib was well tolerated. In fact, in March 2021, he had been treated with sorafenib for 12.5 years, with no sign of recurrence. Analysis of mutations in a previous biopsy revealed a deletion of codon p.I843 D846del located at *PDGFRA* exon 18, leaving intact, the aspartate at codon 842 (D842) in *PDGFRA* exon 18 [141]. The study “Sorafenib in treating patients with a malignant gastrointestinal stromal tumor that progressed during or after previous treatment with imatinib mesylate and sunitinib malate” (ClinicalTrials.gov Id NCT00265798) is a phase II study. The last update was on November 2022. Thirty-eight patients have been enrolled and treated with BAY 43-9006 (sorafenib). The primary endpoint is aimed at evaluating the response rate of GIST patients to the treatment. Secondary endpoints include the evaluation of treatment toxicity, PFS, and OS.

### 7.5. Dasatinib

Dasatinib is a small TKI molecule and a potent inhibitor of BCR-ABL, KIT, and SRC family kinases as well as imatinib-resistant cells. A multicenter, 2-stage phase II trial has been performed in GISTs, producing high metabolic response rates in TKI-naive patients with FDG-PET/CT-positive GIST [142]. Objective tumor response was reached in 25% of patients, including one with an imatinib-resistant mutation in *PDGFRA* exon 18 [143]. In a multicenter clinical trial (Id: NCT02776878), dasatinib was demonstrated active in the treatment of patients with GISTs resistant to imatinib and sunitinib [144].

## 8. Fourth-line Therapy for Resistant GISTs 

The continued development of effective TKIs may further improve the PFS and OS of advanced GISTs (table in below).

### Ripretinib

Ripretinib (Qinlock, DCC-2618) is an orally bioavailable, selective KIT and PDGFRA switch-control inhibitor, active against most *KIT* and *PDGFRA* mutations. Ripretinib binds both to wild-type and mutant forms of KIT and PDGFRA at their switch pocket binding sites, thereby preventing the switch from inactive to active conformations and inactivating their wild-type and mutant forms [145,146]. A phase I clinical trial showed a median PFS of 24 weeks in patients with metastatic GISTs on at least second-line therapy, treated with ripretinib [147]. Ripretinib was also under investigation in a fourth-line randomized phase III trial (INVICTUS, Id: NCT03353753). Patients affected by advanced GISTs resistant to imatinib, sunitinib, and regorafenib, or intolerant to any of these treatments, were enrolled and treated with ripretinib. Ripretinib increased the median PFS compared to the placebo, having an acceptable safety profile [148]. Based on the results from the INVICTUS Study, in May 2020, ripretinib was approved by the FDA for adult patients with advanced GISTs who have received prior treatment with three or more TKIs, including imatinib. The phase III, randomized, open-label study INTRIGUE registered at ClinicalTrials.gov, Id: NCT03673501, enrolled adult patients with advanced GISTs who progressed or had an intolerance to imatinib. The efficacy and safety of ripretinib vs. sunitinib were evaluated. Ripretinib demonstrated good clinical efficacy and lower toxicity in imatinib-resistant advanced GISTs [149]. Very recently, Liu Bo and coworkers described the case of a middle-aged patient whose advanced GIST was resistant to first, second, and third-line therapies. The patient was treated with ripretinib and a partial response was obtained after 6 months [150]. A multicenter phase II, single-arm open-label study of ripretinib aimed to assess efficacy, safety, and pharmacokinetics in patients with advanced GISTs who have progressed on prior antitumor therapies (ClinicalTrials.gov IdNCT04282980). Thirty-nine Chinese patients were enrolled. The results demonstrated that ripretinib can improve the outcomes of patients with advanced GISTs as a drug of fourth- or later-line therapy [151]. The efficacy and safety of ripretinib as a preoperative treatment in locally advanced or recurrent metastatic GISTs resistant to treatment is ongoing. The clinicaltrials.gov Id: NCT05132738, “Ripretinib used for resectable metastatic GISTs after the failure of imatinib therapy” started to enroll patients in 2021 and will be completed in November 2023. Twenty patients will be enrolled, and screening, treatment, and follow-up periods will be assessed for each patient.

The study “Ripretinib combined with surgery in advanced GISTs that have failed imatinib therapy: a multicenter, observational study” (ClinicalTrials.gov Id: NCT05354388), is recruiting patients and will be completed in December 2023. Thirty patients will be enrolled, with a PSF rate at 12 months being the primary outcome, with PFS assessed by radiographic Choi criteria. Xiao and colleagues, by meta-analysis on several databases, analyzed seven randomized controlled trials testing seven different TKIs. Comparing third-line or over-third-line therapies to assess the most active drugs against GISTs, they found that ripretinib is more effective than other drugs with respect to PFS, exhibiting good efficacy, and also for the over-third-line therapy [152].

## 9. New TKIs Targeting GIST Mutations 

The possibility of assessing the GIST genotype and mutations in the *KIT* gene promoted the development of new drugs whose activity is strictly related to GIST genomic mutations.

### 9.1. Bezuclastinib

Bezuclastinib (PLX9486), a selective and orally active c-KIT D816V TKI. PLX9486 targets mutations in exon 17, exerting a selective and potent TKI activity against *KIT* D816V in GISTs, whereas it is less effective with exons 13 and 14-mutated GISTs. In a preclinical study in mice xenografted with GIST tissues from primary as well as secondary resistant tumors harboring mutations in exons 9 and 11, or exons 17 and 18, respectively, PLX9486 revealed good efficacy [153]. A phase I study registered at clinicaltrial.gov Id: NCT02401815, last update posted on May 2021, evaluated the therapeutic activity of PLX9486 alone or in combination with sunitinib or pexidartinib, another TKI active against *KIT* mutations in exons 13 and 14. The study enrolled 36 metastatic GIST patients. A median PFS of >24 weeks with PLX9486 alone was reached. Moreover, a decrease in exon 17 and 18 mutation levels was found in patients who had a clinical benefit after PLX9486 treatment, suggesting that this drug suppresses tumor subclones depending on A-loop mutation for their proliferation. In patients receiving the combination of PLX9486 and sunitinib, a decrease in exon 13 as well as exon 14-mutant alleles were found [154]. More recently, a 1b/2a trial of 39 patients with GISTs evaluated the efficacy of a combination treatment of PLX9486 and sunitinib, whose TKI inhibition targets are different. The combination treatment was well tolerated and clinical benefit for patients was demonstrated [155]. 

### 9.2. Cabozantinib

Cabozantinib is an oral small-molecule multitargeted TKI that may confer an advantage over other TKIs targeting a single receptor (Table 2). In virtue of its ability to target KIT, VEGFR2, MET, and AXL, the FDA-approved cabozantinib treatment for a wide variety of malignancies, including GISTs [156]. In GIST PDX models, cabozantinib showed antitumor activity by inhibiting tumor growth and angiogenesis, either in imatinib-sensitive or imatinib-resistant models [157]. Lu and coworkers demonstrated that cabozantinib is more effective than imatinib against primary *c-KIT* mutations. Moreover, cabozantinib overcame the c*-KIT* gatekeeper T670I mutation and the activation loop mutations causing resistance to imatinib or sunitinib. In vitro and in preclinical in vivo models of KIT-mutated GISTs, cabozantinib showed good efficacy and a long-lasting effect. Further, a dose-dependent antiproliferative efficacy was demonstrated in GIST patients-derived primary cells [158]. Based on its clinical safety and efficacy revealed by a phase II study registered at ClinicalTrials.gov EORTC 1317, Id: NCT02216578, EudraCT 2014-000501-13, cabozantinib could be regarded as a potential drug for metastatic, imatinib and sunitinib-resistant GISTs [159]. 

### 9.3. Avapritinib

Avapritinib (BLU-285) is a potent and selective inhibitor of *PDGFRA* D842V and *KIT* exon 17 mutants [160] (Table 2). In January 2020, avapritinib (AYVAKITTM, Blueprint Medicines Corporation) was the first drug approved by the FDA for the treatment of adults with unresectable or metastatic GISTs harboring the *PDGFRA* D842V mutation on exon 18. A Phase I trial (NAVIGATOR Trial registration Id: NCT02508532) enrolled 231 patients with advanced GISTs or progression on at least second-line TKI therapies, including 56 patients harboring the *PDGFRA* D842V mutation. Preliminary results show a very encouraging 86% response rate in patients with a *PDGFRA* D842V mutation (8 complete responses by RECIST 1.1). The patients on fourth-line therapy or further had response rates of 20–26% [161]. In the updated results, the authors attest to “an unprecedented, durable clinical benefit, with a manageable safety profile”, suggesting the chance to evaluate avapritinib as first-line therapy for patients with advanced GISTs” [162,163]. A very recently published phase III randomized trial (VOYAGER, Id: NCT03465722) compared avapritinib to regorafenib; however, up until now, the primary endpoint (PFS by central radiology per RECIST) has not been reached [164]. Additionally, a comparative retrospective analysis documents more durable survival outcomes in patients with unresectable/metastatic *PDGFRA* D842V-mutant GISTs treated with avapritinib compared with those treated with other TKIs [165].

**Table 2 ijms-24-06026-t002:** Clinical trials evaluating the efficacy of TKIs in imatinib-resistant GISTs.

Line of Therapy	Drug	Study	ID	Phase	Endpoints	Findings	Refs
**2nd**	**Sunitinib**	A study of SU011248 administered on a continuous daily dosing schedule in patients with a gastrointestinal stromal tumor	NCT00137449	II	Safety, pharmacokinetic, and plasma biomarker levels, PR, SD (>24 weeks), PFS, and OS	Efficacy and safety of daily oral administration of 37.5 mg sunitinib for 28 days	[114]
		A study to assess the safety and efficacy of SU11248 in patients with gastrointestinal stromal tumor (GIST)	NCT00075218	III	TTP, PFS, OS, OR, CR, and PR	Good tolerability, longer PD (27.3 weeks) compared to placebo	[116]
**3rd**	**Regorafenib**	A phase II trial of regorafenib in patients with imatinib-resistant gastrointestinal stromal tumor	UMIN000016115	II	PFS rate (>24 weeks), ORR, RR, DCR, and AE	Efficacy in 2nd line therapy for imatinib-resistant GISTs. Secondary mutation in KIT can be predictive of regorafenib efficacy	[128]
		A study of regorafenib as a 3rd-line or beyond treatment for GISTs (GRID)	NCT01271712	III	PSF and OS	Regorafenib PSF > placebo	[129,130]
	**Nilotinib**	A phase II Study aiming to evaluate the efficacy and safety of nilotinib patients with gastrointestinal stromal tumors (GIST) resistant or intolerant to imatinib and or to 2nd-line tyrosine kinas (TK) inhibitor	NCT00633295	II	Efficacy in GISTs resistant to imatinib and/or 2nd-line TKI, safety, and tolerability	Ongoing	
	**Nilotinib**	Phase III Study of nilotinib versus best supportive Care with or without a TKI in patients with gastrointestinal stromal tumors resistant to or intolerant of imatinib and sunitinib	Open-Label Trial	III	Efficacy in patients with advanced GISTs following imatinib and sunitinib failure, and PFS	Longer median OS	[134]
	**Pazopanib**	Efficacy of pazopanib in gastrointestinal stromal tumors (GIST) (PAZOGIST)	NCT01323400	II	OS, PFS, and tolerance profile	Effective on 50% of patients with metastatic or locally advanced GISTs	[136]
	**Sorafenib**	Sorafenib in treating patients with malignant gastrointestinal stromal tumor that progressed during or after previous treatment with imatinib mesylate and sunitinib malate	NCT00265798	II	ORR and OS	Ongoing	
**4th**	**Ripretinib**	Phase 3 study of DCC-2618 vs. placebo in advanced GIST patients who have been treated with prior anticancer therapies (INVICTUS)	NCT03353753	III	Safety profile, PFS, and ORR	Acceptable safety profile and Ripretinib PSF > placebo	[148]
		A study of DCC-2618 vs. sunitinib in advanced GIST patients after treatment with imatinib (INTRIGUE)	NCT03673501	III	PFS and ORR	ORR ripretinib > sunitinib	[149]
		Preoperative treatment of potentially resectable locally advanced and recurrent metastatic GIST after failure of imatinib therapy	NCT05132738	Single arm	NED Rate, surgery rate (proportion of patients who can undergo surgery), and ORR	Ongoing	
		Ripretinib combined with surgery in advanced GIST that have failed imatinib therapy: A multicenter, observational study (NAVIGATOR)	NCT05354388	Observational	PFS (at 12 months), ORR (at 12 months), TTP, and the 2-year overall survival rate	Ongoing	
**NEW TKI Drugs**	**Cabozantinib** (Selective for the T670I mutation)	Ph II CABOGIST in GIST	NCT02216578	II	PFS, OS, and ORR	Potential drug for metastatic, imatinib, and sunitinib-resistant GISTs	[159]
	**Avapritinib** (Selective for the D842V mutation)	Study of BLU-285 in patients with gastrointestinal stromal tumors (GIST) and other relapsed and refractory solid tumors	NCT02508532	I	MTD and ORR	86% response rate by RECIST 1.1 in patients with the *PDGFRA* D842V mutation	[161]
		Study of avapritinib vs. regorafenib in patients with locally advanced unresectable or metastatic GIST (VOYAGER)	NCT03465722	III	PFS, ORR, CR, and OR	The primary endpoint (PFS by central radiology per RECIST) has not been yet reached	[164]

Adverse event (AE); complete response (CR); disease control rate (DCR); maximum tolerated dose (MTD); no evidence of disease (NED); objective response rate (ORR); overall response (OR); overall survival (OS); partial response (PR); progression-free survival (PFS); recurrence-free survival (RFS); response rate (RR); time to progression (TTP); time to tumor response (TTR); evaluation criteria in solid tumors (RECIST).

## 10. BAF Inhibitors and GISTs

BRD9, a subunit of the noncanonical ATP-dependent chromatin remodeling complex ncBAF, has been recently detected in GIST tissues and found to be correlated with tumor size, grade, and progression. BRD-associated proteins are highly dysregulated in tumors, and BRD9 is upregulated in GIST tissues. It has been demonstrated that the downregulation or inhibition of BRD9 could reduce cell proliferation and facilitate PUMA-dependent apoptosis in GISTs. Indeed, the combination of imatinib and GSK602, a potent and selective BRD9 inhibitor, enhances cell apoptosis and reduces cell proliferation via AKT inhibition and PUMA induction. These data suggest a possible future combinational approach for the treatment of GISTs [166].

## 11. Serum Markers for Precision Medicine in GISTs 

The management of GIST patients by precisely targeted therapy need the availability of biomarkers for diagnosis, therapy, and follow-up. In this regard, DNA liquid biopsy, being a noninvasive procedure, may provide a molecular profile of cancer, reflecting the genetic aberrations of cancer cells at a given time [167]. This is the case for primary and secondary *KIT* mutations which have been found in plasma from GIST patients [168,169] and for the extracellular domain of soluble KIT, which is a faithful biomarker of tumor outcome in patients treated with sunitinib after acquired imatinib resistance [170]. In 15 out of 38 plasmas from GIST patients harboring *c-KIT* or *PDGFRA* mutations in tumor tissues, Maier and colleagues identified ctDNA (circulating tumor DNA) bringing tumor-specific mutations, by the allele-specific Taq-Man duplex PCR allele-specific ligation PCR technique. The amount of mutant ctDNA correlated with disease course. In five patients with progressive disease, ctDNA was increased, whereas two negative patients became positive at tumor progression. Based on these findings, mutant ctDNA could be regarded as a tumor-specific biomarker in GISTs [171]. Jilg and colleagues conducted a prospective clinical trial to assess the presence of tumor-specific *c-KIT* and *PDGFRA* mutations in GIST patient plasma. Mutant ctDNA was detected in plasma by allele-specific ligation (L-)PCR and droplet digital (d)PCR. By dPCR analysis, the absolute numbers of ctDNA fragments and the mutant allele frequency were found to strongly correlate with tumor size and response status. (d)PCR was able to detect tumor progression with a specificity of 79.2% and a sensitivity of 55.2%. Moreover, serial ctDNA measurements were performed and found to correlate with individual disease courses. Finally, targeted panel sequencing of four patients identified additional driver mutations in all cases and secondary resistance mutations in two cases, indicating the importance of ctDNA evaluation in the monitoring of GIST patients [172]. An exhaustive description of the most recent and significant biomarkers for GISTs in precision medicine has been made by Yoshiaki Nakamura and collaborators. Interestingly, they highlight the importance of ctDNA genotyping to assess chronological tumor evolution and intratumoral genomic heterogeneity, all necessary for the most accurate selection of patient treatment [173]. These authors also conducted a clinical trial (SCRUM-Japan GOZILA no. UMIN000016343) to validate the utility of ctDNA genotyping compared to tumor tissue sequencing (study GI-SCREEN, 5621 patients) in patients with advanced GISTs. The study demonstrates the importance and sensitivity of ctDNA genotyping in GIST precision medicine. Indeed, ctDNA evaluation reduced the duration of patient enrollment (11 vs. 33 days) and significantly increased the trial enrollment rate (9.5 vs. 4.1%) [174]. In an interesting study, Tun Kiat Ko and coworkers describe a liquid biopsy approach, able to detect primary and secondary mutations in patients with progressive GISTs. They used a customized Archer^®^ LiquidPlex™ targeted panel, enriched by Anchored Multiplex PCR (AMP™). This panel is very sensitive, being able to ligate ctDNA fragments as small as 160 base pairs, generally missed by other platforms, and detect variants at a 0.3% allele frequency with a ctDNA input lower than 1 ng. Plasmas from 46 patients were analyzed for ctDNA mutations. In 7 out of 10 (70%) patients with metastatic GISTs with evidence of disease progression, mutations in ctDNA were found. The analysis of serial plasma samples from six patients with metastatic GISTs after TKI therapy enabled us to ascertain the presence of cDNA mutations acquired at a different time along the progression of the disease. Interestingly, a phosphatidylinositol 3-kinase, catalytic, alpha (*PIK3CA*) c.1633G>A variant at exon 9 in ctDNA was identified and found to correlate with clinical disease progression, as evaluated by computed tomography [175]. These findings fit well with a previous paper by Lasota and colleagues who documented, for the first time, the presence of PIK3CA mutations in GISTS. Analyzing a large cohort of GISTs (529 cases), Lasota and collaborators identified the presence of mutations in the *PIK3CA* gene coding the phosphatidylinositide-3-kinase (PI3K), a downstream KIT signaling pathway effector in only eight high-grade primary and two metastatic GISTs. This finding lets the authors hypothesize that *PIK3CA* mutations may confer a proliferative advantage, becoming dominant in a late stage of the GIST genetic evolution [176]. These findings underline the chance to use these markers, especially PIK3CA, for planning therapy in PD GIST patients. In a preclinical study in mice xenografted with human GISTs, the combination therapy of imatinib plus the PI3K inhibitor GDC-0941 reduced the growth of tumors compared to imatinib alone, conferring a growth advantage for tumor progression in imatinib secondary resistant GISTs [177]. Gupta and collaborators demonstrated that a combination cocktail of drugs targeting KIT/PI3K/MAPK with sunitinib or regorafenib can induce apoptosis in a GIST cell line established from an untreated GIST harboring an IM-sensitive, KIT primary mutation [178]. Moreover, very recently, a clinical study (ClinicalTrials.gov NCT01735968) investigated the safety, tolerability, and efficacy of alpelisib, a phosphatidylinositol 3-kinase inhibitor, in combination with imatinib. Patients with advanced GISTs who had failed in prior therapy with both imatinib and sunitinib were enrolled. Safety and tolerability were acceptable; however, the clinical efficacy was not sufficient to encourage additional clinical testing [179]. 

The importance of serum markers in GISTs is further highlighted by the use of ctDNA to guide or follow-up clinical trials. The phase II trial “ctDNA-guided sunitinib and regorafenib therapy for GIST” (clinicaltrials.gov Id: NCT05366816), starting in 2022, is enrolling GIST patients resistant to first-line imatinib therapy and eventually to second-line therapy with other drugs. The presence/absence of c-*KIT* exon 13 or 17 secondary mutations, determined by ctDNA test in blood, will guide therapeutic choices. Patients harboring *KIT* exon 13 mutations will receive sunitinib and regorafenib at tumor progression. Conversely, patients with *KIT* exon 17 mutations will receive regorafenib and sunitinib at tumor progression. The primary outcome will be the number of participants achieving OR up to 12 months from the treatment, with the estimated end of the study in November 2027. Another phase II trial titled “Ponatinib in patients with metastatic and/or unresectable GISTs following failure or intolerance to prior therapy with imatinib”, (ClinicalTrials.gov Id: NCT03171389) plans to define the efficacy and safety of ponatinib in GIST patients resistant to imatinib, assessing, by liquid biopsies, the presence of circulating DNA from secondary mutations in *KIT* exon 13. The usefulness of liquid biopsies in predicting response to treatment and development of resistance will be also evaluated.

## 12. Conclusions

GISTs are a heterogeneous group of mesenchymal tumors difficult to diagnose and treat. The therapeutic management of GISTs has dramatically improved over the last two decades, mainly due to the discovery of oncogenic drivers and the identification of predictive biomarkers and targeted drugs useful for precision medicine. However, current TKI-based therapies do not satisfy long-term disease control once the disease develops resistance, or because some GIST subtypes do not respond. Hence, the next studies should focus on new targets and drugs, for instance, PI3K and BAF inhibitors. In addition, the next phase of clinical investigations should identify new therapeutic targets, including tumor microenvironment components. In this regard, we foresee that combinations of TKIs and immune checkpoint inhibitors which seem to potentiate traditional TKI-based therapies could lead to a higher response rate and better control of disease progression and survival [63]. 

## Figures and Tables

**Figure 1 ijms-24-06026-f001:**
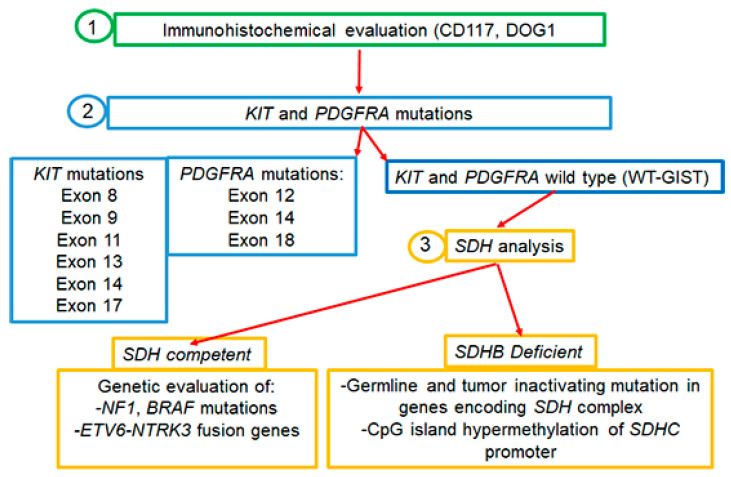
Molecular analysis of GISTs.

**Figure 2 ijms-24-06026-f002:**
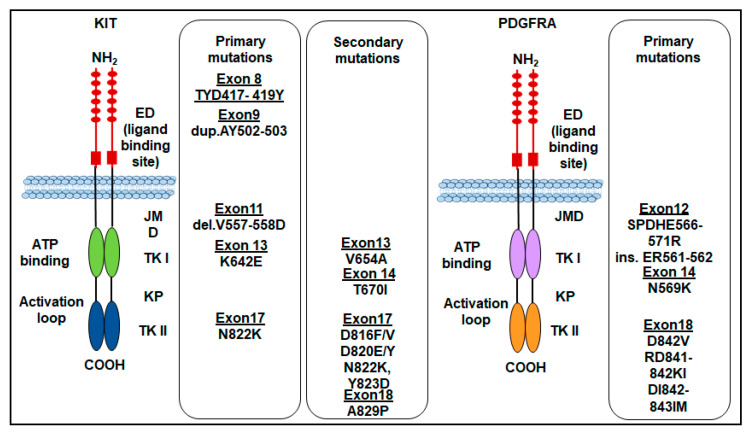
Primary and secondary mutations of *KIT* and *PDGFRA* genes in GISTs. ED: extracellular domain; JMD: juxtamembrane domain; TKI: tyrosine kinase domain 1; TKII: tyrosine kinase domain 2; KP: kinase pocket.

**Table 1 ijms-24-06026-t001:** Molecular subtypes of GISTs and sensitivity to TKIs.

Mutations	Molecular GIST Subtype	Imatinib	Sunitinib	Regorafenib	Ripretinib	Avapritinib
** *KIT* **	Exon 9	S				
	Exon 11	S				
	Exon 13 (V654A)	R	S	R	R	
	Exon 14 (T670I)	R	S	S	S	
	Exon 17 (D816V, D820E, and N822K)	R	R	S/R	S/R	
	Exon 18 (A829P)	R	R	S	S	
** *PDGFRA* **	Exon 12	S	S	S	S	S
	Exon 13	R	R	R	R	
	Exon 14	R	R	R	R	R
	Exon 15	R	R	R	R	R
	Exon 18 (D842V)	R	R	R	R	S
	Exon 18 (Non-D842V)	S	S	S	S	S

Sensitivity (S) or Resistance (R) of KIT and PDGFRA-mutated GISTs to approved drugs.

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
