# Peer review of "Emerging Targeted Therapeutic Strategies to Overcome Imatinib Resistance of Gastrointestinal Stromal Tumors"

_ijms, 2023, doi:10.3390/ijms24076026_

Round 1

Reviewer 1 Report

Dear Editor, 

Please find attached the review report.

Author Response

Responses to Reviewer Comments

Point 1: Please provide some brief information about the Imatinib drug and its resistance mechanism.

Response 1: Informations on Imatinib have been added in the Main text, par. 5. (in RED letters). Senteces describing the mechanisms of resistance to imatinib has been moved from paragraph 5.1 to paragraph 5 (in RED letters). The title of paragraph 5 has been changed.

Point 2: In conclusion (lines 630-632), what do you want to explain?

Response 2: Lines 630-632 have been removed from the text. There had been a simple mistake.

Point 3: Please include abbreviations from all sections after the conclusion.

Response 3: Abbreviations have been added,  after the Conclusions section.

Point 4: Please explain in tabular form the following sections with the use of imatinib drug, study findings, and references.
GIST Kit Mutations
Other GIST Mutations
Imatinib Resistance in GIST
Third-Line Therapy for GISTs That Are Resistant
Resistant GISTs: Fourth-Line Therapy

Response 4: Two table have been added: Table 1. , showing  the sensitivity/resistance of KIT and PDGFRA exons to imatinib and other TKI drugs; Table 2., describing clinical studies, results and references.

Point 5: Please double-check all section writing; some paragraphs are difficult to read.

Response 5: The text has been double checked for Englih language

Note: All the corrections are made in RED to be easily readable.

Reviewer 2 Report

The history of gastrointestinal stromal tumors (GIST) is a nice example how modern biomedical research progress. The identification of this pathological entity, the identification of key molecular oncogenic events, and the development of refined molecular diagnostic tools followed by the use of molecularly targeted pharmacotherapies exemplify how our understanding of a previously obscure form of cancer has improved, and how, based on this improved understanding, our therapeutical options have been enriched, leading to better patient survival. Of course, and unfortunately, the story is not complete, because due to molecular tumor evolution various form of resistance to targeted therapies usually arise, necessitating the development of new drugs. The current paradigm is that the use of such molecules, which in the case of GIST are mostly tyrosine kinase inhibitors, can significantly improve the efficacy of therapy, can overcome resistance and can extend patient survival.

The paper by Masucci et al. reviews GIST molecular subtypes and corresponding molecularly targeted inhibitors with focus on novel molecules currently in early clinical development that show promise for the treatment of GIST resistant to current treatments. This well-written and adequately structured work will help the reader to understand the current state of GIST pharmacotherapy and mechanisms of resistance and will help to familiarize with new drugs currently in early clinical testing. A few minor issues need to be addressed prior publication, as detailed below:

Comments:

Lanes 12-13 : please place e-mail address in a single lane, and delete “-“ prior telephone number.

Lane 44 : “there have been a great debate” : there has been a…

Lanes 63-64 : “it is to pinpoint” : it is to be pointed out that…?

Lane 83 : “recommended the GIST genotyping” : recommended the genotyping of GIST

Lane 83-84 : “since GIST development are” : since the development of GIST is…

Lane 92 : “debated” : discussed?

It would be nice to name the syndromes associated with the occurrence of GIST (such as Carney–Stratakis syndrome).

Lanes 95-96 : “mutations in the KIT region, between the transmembrane and tyrosine kinase, contribute… : …and tyrosine kinase domains, contribute…

Lanes 97-98 and 102 : c-KIT or c-kit?

Lane 111 : “Mutations in the exon 11” : Mutations in exon 11

Lane 111 and elsewhere : “557-558 codon deletion (del V557- D558)” : there seems to be some confusion between the numbering of cDNA (“codon”) and protein (amino acid) here (idem lane 113). Codon and corresponding protein amino acid numbering cannot be numerically the same (see also lanes 130-131 and 339).

Lane 119 : “modification of the extracellular domain of KIT molecule occur” : occurs

Lanes 120-122 : “In these last patients, the drug therapeutic dosage is increased to ameliorate the clinical response [38,39]. So, in patients with exon 9 mutations, the therapeutic dosage of the drug is increased to improve the clinical response [38,39].” : there seems to be some redundancy in the text here.

Lanes 144-145 : “Differential expression of KIT/PDGFRA mutant isoforms have been found in epithelioid and mixed variants of gastric GISTs.” : please explain "differential".

Lanes 146-147 : “the TKR inhibitory regions” : auto-inhibitory regions?

Lane 149 : “juxta-membrane” : juxtamembrane?

Lane 156 : “On the contrary” : on the other hand?

Lane 174 : “Almost 50% WT-GISTs are marked by…” : it would be nice to state that WT is used as WT for KIT and PDGFRA specifically.

Lanes 175-176 : “succinate mitochondrial complex” : succinate dehydrogenase mitochondrial complex

Lane 181 : “GISTs lacking of KIT” : GISTs lacking KIT

Lanes 182-183 : “Their SDH status should be determined since SDH-competent GISTs are aggressive and tend to metastasize” : some clarification is needed because not all SDH-competent GISTs are aggressive.

Lane 186 : “origin” : originate?

Lane 193 : “loss” : lose?

Lane 206 : “targets” : target

Lanes 208-209 : “no partial or complete responses has been obtained” : …have been obtained

Lanes 209-210 : “vandetanib is not effective and not well tolerated in these tumors. Please note that tolerance is related to a patient and not to a tumor.

Lane 224 : “high sensitive” : highly sensitive?

Lanes 239-240 : “a negative regulator of Ras/MAPK and PI3K/mTOR signaling pathways” : Please note that since NF1 is a RasGAP, it is somewhat imprecise to call it a negative regulator of PI3K/mTOR. Probably it would be helpful here to discuss a little RasGAP function and downstream signaling, in order to introduce the notion of negative regulation of the MAPK and PI3K/mTOR pathways by NF1.

Lane 270 : “behaves” ? (performs?)

Lane 273 : “the occurrence of the ETV6-NTRK3 gene fusions have been identified” : the occurrence of … fusions has been identified.

Lanes 281-282 : “These TRK inhibitor” : inhibitors

Lanes 283-284 : “and, actually, phase I and II clinical trials for larotrectinib, and entrectinib (Id: NCT02576431, and Id: NCT02568267, respectively) are ongoing.” : and phase I and II clinical trials for larotrectinib, and entrectinib (Id: NCT02576431, and Id: NCT02568267, respectively) are currently ongoing.

Lanes 291, 386 : “i.e.” : “i.e.:”

Lanes 323-325 and 459 : “The impact of primary and secondary kinase genotype on sunitinib activity was investigated in 97 patients with metastatic, imatinib-resistant/intolerant GISTs” : this is straightforward. Please note, however, that treatment intolerance can occur also in patients with treatment-sensitive tumors, due to drug intolerance by the patient, because of tumor-unrelated (say, for example, cardiac) adverse effects. Therefore, the Reviewer feels that ”resistant” and “intolerant” is not to be used interchangeably in the context of drug resistance in this Paper, and that this should be pointed out somehow in the text.

Lane 336 : Please delete underline.

Lane 339 : “GISTs carrying KITAY502-3 mutations at exon 9 exhibits the highest sensitivity” : GISTs ... exhibit

Lane 354 : “an orally multitarget TKI” : an orally active…

Lane 366 : “PSF” : PFS?

Lane 399 : “nilotinib” : please delete boldface.

Lane 418 : “on” : in?

Lane 421 : “small molecular inhibitor” : small molecular weight inhibitor?

Lanes 427-429 : “Brinch and colleagues describe an advanced GIST patient, 61 yrs old, intolerant to imatinib and progressed after sunitinib and nilotinib treatment. In this patient, sorafenib was well tolerated and on March 2021fs… : years; fs? “on March…” : as of March…?

Lanes 432-435 : “The study “Sorafenib in treating patients with malignant gastrointestinal stromal tumor that progressed during or after previous treatment with imatinib mesylate and sunitinib malate” (ClinicalTrials.gov Id NCT00265798) is a phase II study still active.” : …is a still active phase II study?

Lanes 440-441 : “Dasatinib is a small-TKI molecule and a potent inhibitor of BCR-ABL, KIT, and SRC family kinases as well as imatinib-resistant cells.” : please note that the Bcr/Abl fusion oncoprotein is inhibited also by imatinib. Therefroe, if BCR-ABL is mentioned here, it would maybe appropriate to mention it also earlier, when imatinib (and other drugs that inhibit it) are discussed.

Lanes 443-444 : “in 25% patients” : in 25% of patients.

Lane 447 : “Fourth line Therapy for Resistant GISTs” : why uppercase/lowercase?

Lane 466 : “vs” : vs.

Lanes 478-479 : “In the ClinicalTrials.gov Id: NCT05132738, “Ripretinib used for resectable metastatic GISTs after failure of imatinib therapy”. This phrase is incomplete.

Please reformat lane 481.

Lanes 488-489 : “to assess the more drugs active against GISTs” : ?

Lane 491 : “New TKI Targeting GIST Mutations” : TKIs?

Lanes 492-493 : “The possibility to assess GIST genotype and mutations in KIT gene, promoted the development of new drugs…” : Please delete comma (",").

Lanes 498-499 : “resistant patients” : Please note that it is the tumor that is resistant, not the patient.

Lane 519 : “Lu T and coworkers” : T?

Lane 524 : “ patients derived” : “patient-derived” or : “derived from patients”

Lane 543 : “end point” : endpoint?

Lanes 549-550 : “In this regard, DNA liquid biopsy, being a non -invasive procedure” : non-invasive.

And please note that venipuncture is a (minimally) invasive procedure (although not a surgical procedure).

Lane 556 : “ct” : please explain abbrevaiation (circulating).

Lane 568 : “along the time” : ? Please rephrase.

Lanes 586-587 : “Plasma from 46 patients were analyzed” : was analyzed (or: “plasmas).

Lane 591 : “Phosphatidylinositol” : why uppercase P?

Lane 595 : “mutations in GISTS” why italic?

Lane 608 : exon

Lane 620 : “Therapeutic management of GISTs have dramatically improved” : has improved

Lanes 626-629 : “In this regard, we foresee that combinations of TKIs and immune checkpoint inhibitors which seem to potentiate traditional TKI-based therapies, could lead to a higher response rate, a better control of disease progression and survival.” It would be nice if Authors could add bibliographical references here.

Lanes 630-632 : “Gastrointestinal stromal tumors (GIST)s are soft tissue sarcomas, representing the most common non-epithelial tumors of the gastrointestinal tract. GISTs origin from precursors of the interstitial cells of Cajal, the pacemaker cells, whose function is to signal” (sic). This seems to be a duplicated fragment that should be deleted.

Lane 1110 : “CKIT” : cKIT

Author Response

Responses to Reviewer  Comments

All the comments suggested have been added in the main text. The corrections of English language are in black letters, whereas, whenever phrases/sentences have been added, RED letters have been used to facilitate the reading.

Lanes 626-629: the bibliographical reference has been added

Lanes 630-632 have been removed from the text. There had been a simply mistake.

Reviewer 3 Report

Overall this is a solid review.  The current sections are well-organized and thorough.  I think some small sections can be added in the forefront of next generation therapy. 

Comments

1.       An added section of alternative therapeutic options would be great.  For example, is it beneficial to re-challenge with imatinib for resistant GIST?  Is there a combination therapy with imatinib and other cancer drugs (PI3K/mTOR or others)?

2.       An added section of chromatin-based agent would be great as well (Baf inhibitor?)   

Author Response

Responses to Reviewer  Comments

Point 1: An added section of alternative therapeutic options would be great.  For example, is it beneficial to re-challenge with imatinib for resistant GIST?  Is there a combination therapy with imatinib and other cancer drugs (PI3K/mTOR or others)?

Response 1: As suggested, the combination therapy with imatinib and PI3K/mTOR has been added to the text (paragraph 10, in RED letters), the references have been added.

Point 2: An added section of chromatin-based agent would be great as well (Baf inhibitor?) 

Response 2:  A small paragraph describing initial study with BAF inhibitors in GISTs has been added, as requested (Paragraph 9, in RED LETTERS). 

Reviewer 4 Report

The authors reviewed the trend of therapeutic strategies of Imatinib Resistance of gastrointestinal stromal tumors around kit and PDGFR kinase inhibitions. This perspective well covered current drugs and mentioned the future insight, which is to seek serum markers towards precision medicine development. I think this perspective artcle may be suitable for the publication after minor revision below,

> Even though this topic is to review targets, there are no future target names at all. If the authors identify any potential target names, it would be helpful for the discussion with readers.

Author Response

Responses to Reviewer Comments

Point 1: Even though this topic is to review targets, there are no future target names at all. If the authors identify any potential target names, it would be helpful for the discussion with readers.

Response 1: As requested, some potential focuses for GIST future studying have been briefly indicated in the “Conclusions” section.

Round 2

Reviewer 1 Report

Dear Editor, 

The authors made appropriate modifications, and I would recommend that this review article be published in its entirety.

Dr.BUSA